# Age-Related Blood Levels of Creatine Kinase-MM in Newborns and Patients with Duchenne Muscular Dystrophy: Considerations for the Development of Newborn Screening Algorithms

**DOI:** 10.3390/ijns10020041

**Published:** 2024-06-19

**Authors:** Sarah Nelson Potter, Brooke Migliore, Javan Carter, Veronica R. Copeland, Edward C. Smith, Holly L. Peay, Katerina S. Kucera

**Affiliations:** 1RTI International, Research Triangle Park, Durham, NC 22709, USA; snpotter@rti.org (S.N.P.); bmigliore@rti.org (B.M.); jcarter@rti.org (J.C.); vcopeland@rti.org (V.R.C.); hpeay@rti.org (H.L.P.); 2Department of Pediatrics, Duke University, Durham, NC 27710, USA; edward.smith@duke.edu

**Keywords:** creatine kinase-MM, newborn screening, dried blood spot, neuromuscular disorder, Duchenne muscular dystrophy

## Abstract

Duchenne muscular dystrophy (DMD) is an X-linked progressive disorder and the most common type of muscular dystrophy in children. As newborn screening (NBS) for DMD undergoes evaluation for the Recommended Uniform Screening Panel and is already mandated in multiple states, refining NBS algorithms is of utmost importance. NBS for DMD involves measuring creatine kinase-MM (CK-MM) concentration—a biomarker of muscle damage—in dried blood spots. The current test is FDA-approved for samples obtained less than 72 h after birth. Separate reference ranges are needed for samples collected later than 72 h after birth. In this study, we investigated the relationship between age and CK-MM in presumed healthy newborns to inform NBS algorithm designs. In patients with DMD, CK-MM is persistently elevated in childhood and adolescence, while it may be transiently elevated for other reasons in healthy newborns. CK-MM decrease over time was demonstrated by a population sample of 20,306 presumed healthy newborns tested between 0 and 60 days of life and repeat testing of 53 newborns on two separate days. In the population sample, CK-MM concentration was highest in the second 12 h period of life (median = 318 ng/mL) when only 57.6% of newborns tested below 360 ng/mL, the lowest previously published cutoff. By 72 h of age, median CK-MM concentration was 97 ng/mL, and 96.0% of infants had concentrations below 360 ng/mL. Between 72 h and 60 days, median CK-MM concentration ranged from 32 to 37 ng/mL. Establishing age-related cutoffs is crucial for optimizing the sensitivity and specificity of NBS for DMD.

## 1. Introduction

Creatine kinase (CK) catalyzes the conversion of creatine to phosphocreatine, which is used to regenerate adenosine triphosphate in cells. CK is particularly concentrated in tissues with high energy demand. Cellular damage in these tissues can cause CK to leak from cells to blood serum; therefore, elevated serum CK is a secondary biomarker of damage in tissues such as the heart, skeletal muscle, kidneys, and brain.

CK is a dimeric molecule composed of subunits M and B. The combination of these subunits results in three tissue-specific isoenzymes: -MM, -MB, and -BB. The CK-MM isoenzyme is the most abundant isoform in skeletal muscle, accounting for 98% of total skeletal muscle CK. Transient CK-MM increase in blood has been associated with trauma, myocardial infarction, polymyositis, stroke, cerebral disease, and exercise [1], while sustained CK-MM elevation is a biomarker of neuromuscular disease [2,3].

Duchenne muscular dystrophy (DMD) is the most common degenerative neuromuscular disorder. DMD is caused by pathogenic variants in the X-linked *DMD* gene that lead to the absence of the protein dystrophin, which causes deterioration of structural stability in muscle cells [4]. Symptoms of DMD typically appear between 2 and 3 years of age, with a mean age of diagnosis around 5 years [5,6]. DMD eventually leads to progressive and severe muscle wasting and premature death. The incidence of DMD is approximately 1 in 5000 live male births [7]. Females are typically asymptomatic carriers or have mild symptomology. Female cases are exceedingly rare [8]. However, manifesting females, including those with disease severity comparable to males, exist as a result of X chromosome monosomy in individuals with Turner syndrome and bi-allelic inherited and de novo pathogenic variants or as a consequence of non-random X chromosome inactivation. In such cases, the X-chromosome with the unaffected copy of *DMD* is inactivated in most or all cells, resulting in insufficient production of the DMD protein. Additionally, up to approximately 19% of carriers develop skeletal muscle symptoms, and up to approximately 17% develop cardiomyopathy [9]. Given that symptomatic females exist, it is important to identify them early and ensure equitable access to early treatments and services.

The detection of CK-MM in newborn dried blood spots (DBSs) by the recently developed isoform-specific fluoroimmunoassay [#3311-001U, Revvity] has improved the specificity of screening compared to previous screening for total CK (all isoforms, namely -MM, -MB, and -BB, combined). However, because CK-MM is an indirect marker of DMD and can also be elevated because of other causes of muscle damage [10,11,12,13,14,15,16,17,18,19,20,21,22], the risk for false positives remains. Indeed, recent studies have shown that transiently elevated CK-MM in the early newborn period could be a significant cause of false-positive NBS referrals. This is because, for most newborns, NBS DBSs are collected in the first 24 to 48 h of life when elevated CK-MM resulting from traumatic birth events (e.g., the use of forceps, vacuum extraction, fractures, or a nuchal cord) may still be returning to normal [23]. The level of CK-MM in blood at the time of typical NBS may therefore be indistinguishable among newborns with DMD and those recovering from birth trauma [21,24]. The transient nature of nongenetic hyperCKemia thus presents an opportunity for implementing repeated CK-MM testing to differentiate newborns with DMD experiencing sustained hyperCKemia and to minimize false-positive referrals for newborns whose CK-MM levels normalize within the first few days of life [10,14,16,21].

Most NBS programs use a one-screen model where newborns are only screened at birth, typically between 24 and 48 h of life. However, established repeat testing mechanisms can be utilized in the development of NBS algorithms for DMD. For example, 12 U.S. states use a two-screen model where all newborns are screened at birth and then a second time at 1 to 2 weeks of age, often during a “well-baby” visit [25]. In addition, repeat specimen collection and testing are used in either scenario if the initial specimen did not meet quality control criteria (e.g., poor quality or collected before 24 h of life) or after an initial abnormal or borderline result.

The Genetic Screening Processor (GSP) Neonatal Creatine KinaseMM kit is FDA-approved for use from birth to 72 h of age, and specimen collection and handling requirements indicate that the DBS collection timeframe should occur 24 to 72 h after birth [#3311-001U, Revvity]. Therefore, each laboratory considering repeat specimen testing as a part of the NBS algorithm will need to evaluate its use outside of this age range. In addition, *DMD* sequencing is another effective strategy that may be considered in algorithm development to increase specificity. Incorporating *DMD* sequencing into NBS algorithms has been discussed elsewhere [12,26].

Multiple NBS pilot studies have reported various age-related CK-MM cutoffs. Some of these studies have also determined sex, birthweight (BW), and gestational term-related (i.e., preterm vs. full-term) cutoffs (Table 1 and Figure 1). The New York State (NYS) NBS and RTI International’s Early Check NBS voluntary research programs have both implemented cutoffs for CK-MM that decrease as the age at sample collection increases [12,16,17,20,21]. NYS set both borderline and referral cutoffs; new specimens were requested for any borderline samples. Early Check also established cutoffs based on BW given that BW and gestational age are positively correlated with CK-MM levels [16,17,20,22]. Additionally, National Taiwan University Hospital implemented separate cutoffs for full-term (750 ng/mL) versus preterm (650 ng/mL) newborns [10], whereas the supplemental DMD NBS program at Brigham and Women’s Hospital, which only included full-term newborns, had separate cutoffs for males (1080 ng/mL) and females (958 ng/mL) [19]. Two studies in China only included males [13,14], and multiple studies have found that CK-MM levels are slightly higher in males compared to females [16,17,27]. However, despite the X-linked nature of DMD, female carriers can be symptomatic, and given recent improvements in testing and interventions, implementing NBS for DMD for both males and females is beneficial [28]. Additionally, with the exception of Parad et al. (2021), who relied on the previously established cutoffs for males and females (Revvity), other recent studies have implemented cutoffs between the 99th and 99.985th percentiles based on study-specific population distributions [19]. Generally, past studies have demonstrated that in unaffected newborns, CK-MM levels are, on average, higher at birth and decrease (i.e., stabilize) at approximately 1 week of life or sooner [17,20].

NBS for DMD is currently being evaluated for the Recommended Uniform Screening Panel in the United States and has been mandated in several states, including Minnesota, New York, and Ohio. Internationally, Taiwan is the only country that routinely screens for DMD as a part of population-based NBS [10,29]. Population data are still lacking in various age groups to fully refine reference intervals and screening algorithms. In this study, we investigated the relationship between age and CK-MM levels in newborns and in patients with DMD and defined reference intervals beyond the FDA-approved GSP Neonatal CK-MM kit to help inform NBS algorithms for DMD.

## 2. Materials and Methods

### 2.1. Specimens

Newborn male and female residual DBSs from NBS at the North Carolina State Laboratory of Public Health were stored in airtight, desiccated bins held at room temperature until testing. Between November 2020 and July 2023, both deidentified specimens and specimens from the Early Check NBS research study were included either as singletons or as paired DBSs from the same individual at different ages (*n* = 53, initial specimen age range (0, 68) hours, repeat specimen age range (38, 656) hours) [16,18]. Additional DBSs were created as previously described [30] from venous blood (EDTA) from 15 male patients with DMD (ages 9–25 years) followed at Duke Children’s Neuromuscular Clinic.

### 2.2. Testing Instrument and Assay Kit

Testing was performed on one GSP (#2021-0010) using the FDA-approved GSP Neonatal CK-MM kit as previously described [24]. DBSs were punched (3.2 mm) using a DBS puncher (Perkin Elmer/Revvity: Part Number: 1296-071) in singlicate into 96-well test plates containing controls and calibrators as per the kit insert [Revvity Cat. No 3311-001U]. CK-MM levels were quantified through the reportable range of 29.2–8000 ng/mL. Values with results < 29.2 and >8000 ng/mL were recorded for a subset of specimens using a configuration in the GSP software (version 12-0-4-2), with the understanding that they are not accurate because they lie outside of the linear range of the assay.

### 2.3. Statistical Analyses

Data analysis was performed using Stata, R (version 4.3.1, 16 June 2023), and RStudio (v 2023.12.1+402). Descriptive summaries and tabulations were performed using Stata MP/17.0. Plots were created in R with ggplot2 (version 3.4.2). Linear regression equations, correlation coefficients, and plots with 95% confidence intervals were generated in R using the ggplot2 geom_smooth lm and stat_regline_equation functions. For samples at the lower limit of linearity of the assay (<29.2 ng/mL), the midpoint between 0 and 29.2 ng/mL (14.6 ng/mL) was used for statistical analyses as previously described by Kucera et al. (2024) [16].

## 3. Results

### 3.1. CK and CK-MM Levels in Patients with Duchenne Muscular Dystrophy

Total CK and CK-MM levels were elevated in childhood and into teenage years in the 15 patients with DMD (Figure 2). Total CK and CK-MM levels were strongly correlated (R2 = 0.89). Both total CK and CK-MM levels were negatively correlated with age, presumably because of severe muscle loss [31,32].

### 3.2. CK-MM Levels in Paired Initial and Repeat Newborn Specimens

CK-MM concentration in paired specimens collected on different days from each of the 53 newborns who were presumed healthy indicated rapid CK-MM normalization soon after birth for individuals with initially elevated CK-MM levels. In most newborns (94.3%), CK-MM concentration decreased over time or remained at or below the lower limit of the reportable range (29.2 ng/mL) (Figure 3). Only three individuals (5.6%) experienced a CK-MM increase. Two of those cases had a marginal increase: one increased from 192 to 208 ng/mL (8.3% increase) between 49 and 66 h old, and the other increased from 87 to 94 ng/mL (8.0% increase) between 25 and 109 h. One case had an unexplained CK-MM increase from 147 to 382 ng/mL (159.9% increase) between 33 and 209 h.

### 3.3. Age-Based CK-MM Ranges in the Newborn Period

CK-MM screening results were analyzed for groups within and outside of the kit specifications (Figure 4, Table 2). The first 2 days are displayed in 12 h increments, illustrating the relatively rapid CK-MM concentration change in the early newborn period. The remaining groups are displayed in day ranges up to 60 days of life. The red-filled boxes highlight the distributions of CK-MM levels in the groups of newborns for whom DBSs were collected after 72 h of age, which is outside of the FDA-approved kit specifications. CK-MM concentrations do not follow a normal distribution; therefore, medians and percentiles were compared.

On average, newborns who are tested in the first 2 days of life have higher levels of CK-MM concentration than those who are tested later. By 49 to 72 h/day 3, median CK-MM concentration was 97 ng/mL, and 96.0% of newborns had concentrations below 360 ng/mL. The percentile analysis and assessment of the proportion of the sample below the lowest previously used cutoff (360 ng/mL) [16] indicated normalization of CK-MM concentration to baseline from day 4 to day 10. After day 10, over 99% of CK-MM values were below 360 ng/mL. Between days 11 and 60, the mean and median CK-MM values were closely aligned, and the distributions in these groups were highly right-skewed with only seven newborns (0.58%) having CK-MM concentrations above 360 ng/mL.

## 4. Discussion

We investigated the relationship between age and CK-MM levels in newborns and defined reference intervals beyond the FDA-approved GSP Neonatal CK-MM kit. As expected, both total CK and CK-MM levels in patients with DMD remained elevated above the levels observed in the general population during childhood and adolescence. Both CK and CK-MM were found to be negatively correlated with age.

Conversely, the rapid stabilization of blood CK-MM in healthy newborns can be readily detected with repeat testing within a few days after birth. Repeat specimens were not available from newborns with DMD for comparison; however, given the sustained CK-MM elevation in older children, and previously reported cases of repeat testing in newborns with neuromuscular conditions [10,14,21], repeat CK-MM testing is expected to distinguish DMD cases from unaffected newborns and significantly improve the positive predictive value of NBS.

A considerable proportion of newborns screened in the first 3 days after birth, when NBS typically occurs, had higher levels of CK-MM relative to the newborns who were screened after 72 h of age, indicating that age-based CK-MM levels must be considered during assay verification and evaluation of options for appropriate NBS algorithms. Multiple cutoffs have been previously proposed by different NBS programs based on variables, including age [11]. Considering age in NBS algorithm design is significant to optimize both positive and negative predictive values.

Most newborns undergo NBS within the first week of life; however, some newborns may not be tested for several weeks. Furthermore, for programs that implement repeat NBS, tests are performed after the 72 h period that is currently covered by FDA approval for the commonly used GSP Neonatal CK-MM kit. The temporal profile of the CK-MM reference intervals beyond 72 h of age informs about appropriate cutoffs for newborns who may receive NBS at an older age and about the appropriate timing of second specimen collection for repeat testing that will reflect the expected CK-MM normalization to baseline.

We previously compared CK-MM levels in newborn males and females and determined that the population difference does not warrant separate NBS cutoffs for these groups [16]. Females identified through CK-MM-based NBS are expected to have significant muscle damage, resulting in CK-MM values above the NBS threshold, while CK-MM levels in most asymptomatic females may be indistinguishable from the general population. The early identification of symptomatic females and females at risk of becoming symptomatic will improve equitable access to treatments and services; however, evidence is needed to determine the rate of identification of symptomatic and asymptomatic female cases and carriers using CK-MM-based NBS and the benefits of early intervention for identified symptomatic females.

Multiple algorithms involving repeat CK-MM testing and *DMD* sequencing have been piloted and considered for NBS implementation. Repeat CK-MM testing and targeted genomic sequencing of the *DMD* gene are important strategies to consider for NBS algorithms to increase testing specificity. However, it has yet to be determined how feasible and time-effective NBS algorithms that involve or do not involve sequencing and repeat CK-MM will be. Additionally, residual clinical sensitivity limitations remain with respect to *DMD* sequencing; therefore, a subset of DMD cases may test positive with CK-MM screening but will not be detected by current targeted next-generation sequencing tests [33,34,35]. Repeat CK-MM testing may reduce the need for sequencing; however, a new specimen procurement for repeat CK-MM testing will incur a separate cost and may delay diagnosis. Considerations around program-specific facilitators and barriers (e.g., differences in state specimen collection requirements [one- vs. two-screen states], newborn age requirements for first and second DBS collection, newborn population demographics, accessibility to in-house or outsourced *DMD* sequencing, and historical rates of repeat specimen collection) will be important in decision making as NBS programs prepare to implement DMD screening for their specific populations.

## 5. Conclusions

Factors unrelated to DMD (e.g., birth trauma) can affect CK-MM levels in the newborn period and lead to transiently elevated CK-MM levels detected in DBSs collected within the first days of life for NBS. This may interfere with differentiating healthy newborns from newborns with DMD, who are characterized by sustained CK-MM elevation at birth, during childhood, and during the teenage years. Many previous studies have developed age-specific cutoffs to optimize the sensitivity and specificity of NBS for DMD with the GSP Neonatal CK-MM kit that are informative for public health implementation options. The results from this study provide guidance for the use of the FDA-approved GSP Neonatal CK-MM kit beyond 72 h of age. Each program needs to consider population- and program-specific factors when designing their algorithms, including the minimum age of DBS collection at which transiently elevated CK-MM has normalized. We show that collection of repeat DBSs should occur at least after 72 h of age, when 96% of newborns have CK-MM levels below the lowest previously used cutoff (i.e., 360 ng/mL) [16]. By 10 days of life, CK-MM normalizes for over 99% of newborns.

## Figures and Tables

**Figure 1 IJNS-10-00041-f001:**
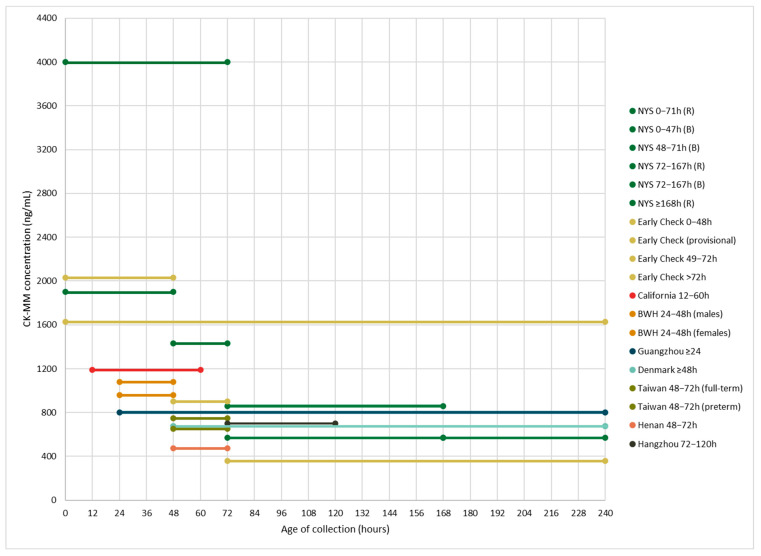
Age-related creatine kinase-MM cutoff levels in newborn screening pilot studies. Although the x-axis ends at 240 h (i.e., 10 days), four groups did not have upper limits on their age at collection cutoffs: NYS, Early Check/RTI, the Danish Neonatal Screening Biobank, and Guangzhou NBS Center, at ≥168, >72, ≥48, and ≥24 h, respectively. In the other studies, the time frames indicated were based on the times of sample collection. Refer to Table 1 for the full names of NBS pilot studies and additional information.

**Figure 2 IJNS-10-00041-f002:**
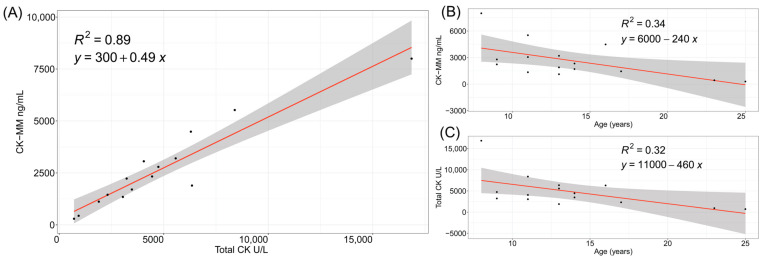
Total CK and CK-MM levels are correlated in patients with DMD. (**A**) Total CK and CK-MM scatter plot and correlation. (**B**) CK-MM correlation with age in years and (**C**) total CK correlation with age in years. Gray shading outside of trendlines indicates 95% confidence intervals.

**Figure 3 IJNS-10-00041-f003:**
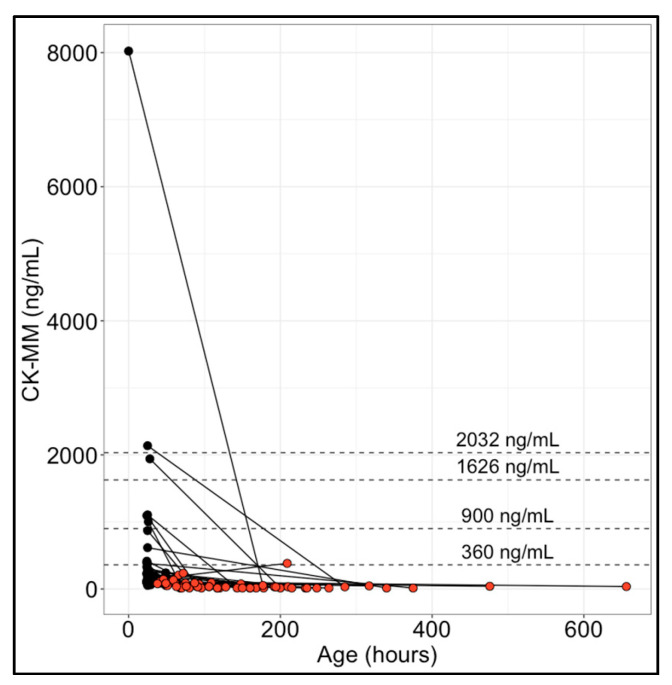
CK-MM levels in paired specimens from 53 newborns. The initial specimen results (black dots) and repeat specimen results (red-filled dots) collected from the same newborn on different days are connected by solid black lines. Example cutoffs (horizontal dashed lines) from a previous study [16] are shown for reference.

**Figure 4 IJNS-10-00041-f004:**
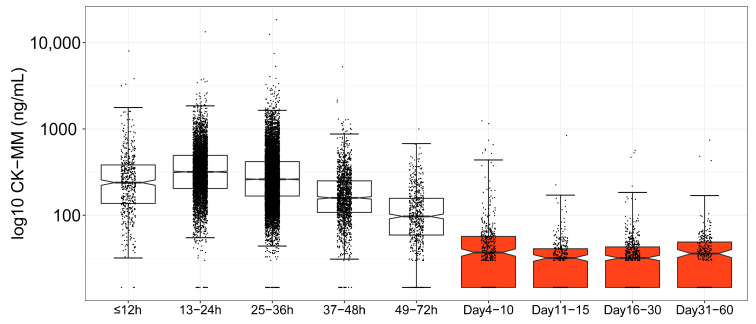
Log10 CK-MM concentration by hour/day range. The box and whisker plots for each age range within (white fill) and outside (red fill) the kit specifications represent the inner quartile ranges (i.e., 50% of the data distributions in each group) with the median indicated by a horizontal line. The whiskers represent the upper and lower quartiles, and the notches show differences in the medians. The midpoint between 0 and 29.2 ng/mL (14.6 ng/mL) was used for values below the lower limit of the reportable range of the assay (<29.2 ng/mL). Values above the upper limit of the reportable range (>8000 ng/mL) are included but not considered quantitative.

**Table 1 IJNS-10-00041-t001:** Age-related creatine kinase-MM cutoffs in newborn screening pilot studies.

Population and Publication(s)	*N*	Age at Collection Range (Hours) and CK-MM Cutoffs (ng/mL)	Percentile Cutoff
New York State NBS program ^1^ Hartnett et al. (2022)—[12] Maloney et al. (2023)—[17] Park et al. (2022)—[20] Tavakoli et al. (2023)—[21]	36,781	0–47 h: 1990 (B), 4000 (R) 48–71 h: 1430 (B), 4000 (R) 72–167 h: 571 (B), 860 (R) ≥168 h: 571 (R)	99.5th
Early Check NBS voluntary research study (RTI International) ^2^ Kucera et al. (2024)—[16] Migliore et al. (2022)—[18]	13,354	Provisional cutoff: 1626 0–48 h (BW: >1500 g): 2032 49–72 h: 900 >72 h (BW: ≤1500 g): 360	99.5th
California Biobank Program and Danish Neonatal Screening Biobank Timonen et al. (2019)—[22]	California (CA): 719 Denmark (DK): 1422	12–60 h (CA): 1190 ≥48 h (DK): 675	CA: 99th DK: 99.5th
Guangzhou NBS Center (Guangzhou City, China) Jia et al. (2022)—[13]	62,553	≥24: 800	99.7th
Supplemental DMD NBS Program (Brigham and Women’s Hospital, Boston, Massachusetts) ^3^ Parad et al. (2021)—[19]	1379	24–48 h: 1080 (males), 958 (females)	N/A
NBS Center at National Taiwan University Hospital Chien et al. (2022)—[10]	50,572	48–72 h: 750 (full-term), 650 (preterm)	99th
NBS Center of Henan (Henan Province, China) ^4^ Jia et al. (2023)—[14]	13,110	48–72 h: 472	99.8th
Hangzhou NBS program (Zhejiang Province, China) ^4^ Ke et al. (2017)—[15]	18,424	72–120 h: 700	99.985th

*Note*. CK-MM: creatine kinase-MM. NBS: newborn screening. ^1^ Sample size for 2-year pilot study; B: borderline cutoff (new specimen requested for repeat screening); R: referral cutoff; additional specimen requested for samples collected <24 h; percentile cutoff for borderline range. ^2^ BW: birthweight cutoffs; a provisional cutoff of 1626 ng/mL (based on 99.5th percentile) was used at the beginning of the study. ^3^ Full-term newborns only. ^4^ Males only.

**Table 2 IJNS-10-00041-t002:** CK-MM concentration by hour/day range (*N* = 20,306).

						Percentiles			% Below 360 ng/mL
Group	*N*	Mean	Median	Max	*SD*	99.0th	99.5th	99.75th	
≤12 h	499	327	238	3830	360	619	3241	3698	73.1
13–24 h	5167	394	318	3799	298	721	1857	2253	57.6
25–36 h	10,776	338	261	7513	290	638	1699	2232	67.8
37–48 h	1595	208	159	5268	212	371	1159	1309	89.0
49–72 h	577	126	97	999	104	245	668	855	96.0
Days 4–10	477	58	37	1242	108	86	991	1224	97.5
Days 11–15	341	35	32	839	50	55	403	839	99.7
Days 16–30	563	36	32	564	43	57	480	550	99.5
Days 31–60	311	42	36	741	56	61	596	741	99.0

*Note*. The lower limit of the reportable CK-MM range is 29.2 ng/mL; this value was found in each group. Four CK-MM values above the 8000 ng/mL upper limit of the reportable range were excluded from this analysis.

## Data Availability

The data are not publicly available due to privacy restrictions.

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
