# Peer review of "Age-Related Blood Levels of Creatine Kinase-MM in Newborns and Patients with Duchenne Muscular Dystrophy: Considerations for the Development of Newborn Screening Algorithms"

_2409-515X, 2024, doi:10.3390/ijns10020041_

Round 1

Reviewer 1 Report

Comments and Suggestions for Authors

Dear authors,

I very much appreciated the chance to review your manuscript. The scope of your study is well described, and clearly presented. However, there are still a few points minor that need clarification.

1. With DMD being an x-linked disorders, you are refer to 2 publications [8] and [26]. Although in both papers it is described that female carriers can develope symptoms, you argue in lines 50-51 that this is extremely rare, and carriers have only mild symptoms, and lines 97-99 argue (on the basis of [26]), that carriers can be symptomatic, and therefore NBS also in females is beneficial. This discrepancy should be adjusted, and this part of the introduction needs revision.

2. A discussion whether females should be screened or not is totally missing, but this a crucial decision for DMD-NBS. Since carrier status for DMD is, if at all, only a late onset disease, and early detection has no benefit for the carriers. It seems questionable, whether there is real benefit to detect female newborn carriers, and therefore strictly complying to the Wilson and Jungner criteria would exclude DMD-NBS in females. In addition with every boy with DMD, there is anyway one female carrier detected (the mother). Every NBS program that started with DMD-NBS should include the clinical situation of the mother. This could then result in a proper decision for screening females. These points should be discussed in detail.

3. You should choose a unique time frame for the blood draw for NBS. It is mostly 24 to 72 hours, but in line 68 you change to 24 to 48 hours.

4. Concerning the FDA approval only from birth to 72 hours of age. I presume this restriction is solely based on the fact the PE/Revvity just provided data over this time periode. Your data clearly that using age related cut-offs, this text kit can be used also for older newborns. I would encourage you to include this in the discussion.

5. Figure 1 is extremely hard to understand. These data would be much better understandable, when the data are presented in a table.

6. In figure 4 you use different colours for the samples taken up 72 hours (white) and after that (red). Since CK-MM is the same enzyme and does not change. (see also point 4)

7. In line 250 you describe one reason for elevated CK could be birth trauma. It would be interesting to compare the CK-MM values from babies that were normaly delivered, and those the were delivered by cesarian section. In case that these data are available, you should include them, if not it should be at least discussed, and probably integrated in further studies.

Reviewer 2 Report

Comments and Suggestions for Authors

I have read this paper with great interest, and a background on neonatology and clinical research, including screening practices in early neonatal life.

I agree that ‘preparedness’ to initiate Duchenne screening is relevant, but neonatal screening for a disease with limited immediate impact during infancy, as well as during childhood (presymptomatic versus symptomatic) is not fully compliant with the current guidelines on screening practices. Although I’m very well aware of the push to screen in an attempt to facilitate early presymptomatic trials, this should be weighted to individual interests of families and their infants

A specific issue for a neonatologist is what we mean with ‘presumed healthy’ newborns. Is there anything known on eg. CK values after instrumental delivery, or perinatal asphyxia, or in essence who and how many cases were excluded in the studies mentioned for clinical conditions. In your conclusion, you refer to birth trauma, but we do need more detailed information on the type of events, exclusion criteria and the portion of cases excluded because of exclusion criteria.

How does intramuscular vitamin K administered at birth affect these data (likely explaining the age dependent pattern) ? In essence, I do think that additional ‘circumstantial’ information besides the postnatal age matters, like eg centers with oral versus iv vitamin K prophylaxis.

Were reference values only constructed in male infants ? and have you assess the impact of sex on your data (although likely not fully powered to do so, but as secondary analysis)
